# On the Efficiency of a Lightweight Authentication and Privacy Preservation Scheme for MQTT

**Sijia Tian** and **Vassilios G. Vassilakis** *

Department of Computer Science, University of York, York YO10 5GH, UK; st1331@york.ac.uk
* Correspondence: vv573@york.ac.uk

**Abstract:** The Internet of Things (IoT) deployment in emerging markets has increased dramatically, making security a prominent issue in IoT communication. Several protocols are available for IoT communication; among them, Message Queuing Telemetry Transport (MQTT) is pervasive in intelligent applications. However, MQTT is designed for resource-constrained IoT devices and, by default, does not have a security scheme, necessitating an additional security scheme to overcome its weaknesses. The security vulnerabilities in MQTT inherently lead to overhead and poor communication performance. Adding a lightweight security framework for MQTT is essential to overcome these problems in a resource-constrained environment. The conventional MQTT security schemes present a single trusted scheme and perform attribute verification and key generation, which tend to be a bottleneck at the server and pave the way for various security attacks. In addition to that, using the same secret key for an extended period and a flawed key revocation system can affect the security of MQTT. To address these issues, we propose an Improved Ciphertext Policy-Attribute-Based Encryption (ICP-ABE) integrated with a lightweight symmetric encryption scheme, PRESENT, to improve the security of MQTT. In this work, the PRESENT algorithm enables the secure sharing of blind keys among clients. We evaluated a previously proposed ICP-ABE scheme from the perspective of energy consumption and communication overhead. Furthermore, we evaluated the efficiency of the scheme using provable security and formal methods. The simulation results showed that the proposed scheme consumes less energy in standard and attack scenarios than the simple PRESENT, Key Schedule Algorithm (KSA)-PRESENT Secure Message Queue Telemetry Transport (SMQTT), and ECC-RSA frameworks, with a topology of 30 nodes. In general, the proposed lightweight security framework for MQTT addresses the vulnerabilities of MQTT and ensures secure communication in a resource-constrained environment, making it a promising solution for IoT applications in emerging markets.

**Keywords:** MQTT; lightweight authentication; ICP-ABE; PRESENT; Cooja



## 1. Introduction

According to the latest "State of the IoT 2022" research report from IoT Analytics, the Internet of Things (IoT) market is expected to grow by 18% to reach 14.4 billion active connections by 2022 [1]. The operation of a large IoT network poses numerous hurdles, including device authorisation, safeguarding measures, and platform service infrastructure, which could burden network bandwidth and communication protocols [2–4]. The Message Queuing Telemetry Transport (MQTT) protocol has become the preferred protocol for the IoT community due to its lightweight and efficient features, reliable messaging, and extensive connectivity support. With the notable increase in the number of IoT devices, MQTT has faced particular challenges, including security vulnerabilities. The critical security issues that MQTT must address are authentication and privacy [5–7]. The broker-based publish/subscribe model makes MQTT communications more vulnerable, as malicious nodes can spoof the identities of legitimate publishers and communication participants by sending unauthorised fake data.

Authentication has been integrated with key update mechanisms in multiple research studies to enhance MQTT security [7–11]. However, frequent key updates can degrade the performance of MQTT in terms of overhead and energy. Therefore, it is essential to design lightweight and efficient security mechanisms for MQTT to address security concerns without compromising performance. In general, the growth of IoT devices has introduced significant challenges to communication protocols, communication security, and the platform service architecture [12,13]. As the preferred protocol for IoT communication, MQTT must address the security concerns arising from its broker-based publish/subscribe model. The proposed security mechanisms for MQTT must be lightweight and efficient, ensuring secure and reliable communication without compromising performance.

In IoT environments, lightweight authentication is crucial to enhance the security of MQTT without affecting its performance. Lightweight authentication uses the self-key update and authentication scheme [14]. Attribute-Based Encryption (ABE) [15] is commonly used in IoT solutions. However, traditional access control solutions require encryption of IoT data using the Ciphertext Policy-Attribute-Based Encryption (CP-ABE) scheme, and only users who conform to the access control policy can decrypt the encrypted information received [16,17]. ABE schemes usually involve high computational complexity and are challenging to implement in resource-limited wireless sensors with limited power and computational capacity. Therefore, a lightweight MQTT authentication and privacy-preserving scheme for publish/subscribe communication models must be proposed.

In this study, we enhanced the MQTT protocol's security by introducing a lightweight Improved Ciphertext Policy-Attribute-Based Encryption (ICP-ABE) [18] strategy. The main contributions of the proposed scheme are as follows:

1. We implemented a lightweight authentication mechanism using the PRESENT [19] algorithm and an improved CP-ABE [20] scheme to ensure secure MQTT communication between IoT devices while maintaining optimal performance.
2. To reduce the computational overhead of the ABE solution, we separated the attribute-auditing process from the secret-key-generation process without revealing the privacy level of the IoT users.
3. We permitted the IoT devices to select a keep-alive parameter and revoke the keys automatically. Thus, it gives protection against slow Denial of Service (DoS) attacks with minimum energy consumption and overhead.
4. We utilized a self-key revocation scheme and instructed MQTT clients to manage the key storage process with minimum cost and overhead without affecting the performance level of MQTT.
5. We provided mathematical proofs for the cryptographic properties of the ICP-ABE security framework using provable security.
6. We analysed the energy consumption and communication overheads of ICP-ABE using the Cooja simulator.

The rest of the paper is organised as follows. Section 2 discusses existing MQTT authentication schemes and highlights their limitations. In Section 3, we describe the details of our proposed scheme. In Section 4, we analyse the security and computational costs of the ICP-ABE scheme. Section 5 provides simulation results that compare our scheme with two existing CP-ABE-based schemes. Section 6 concludes the paper.

## 2. Related Work

Some existing security schemes rely on the Transport Layer Security (TLS) to secure MQTT communications [21]. TLS is widely used due to its security provisioning, ease of implementation, and interoperability with various systems. However, the high complexity processes in TLS, such as handshake message overhead and certificate management, make it inadequate for MQTT, in particular for resource-constrained devices [22]. Therefore, it is necessary to implement lightweight encryption schemes for MQTT. To address this, Park et al. [7] designed a new protocol, MQTT-SN, to bootstrap MQTT security. MQTT-SN supports fine-grained access control and secure communication by establishing a

direct secure channel between a publisher and its corresponding subscribers through the use of a topic certificate. In particular, MQTT-SN integrates essential security-related functions about mutual authentication and connection establishment into the standard MQTT protocol. However, the connection is established only when the membership of a group is changed, and re-keying is used to revoke subscribers from groups, which may give an attacker enough time to trace the secret key. Moreover, this solution uses Elliptic Curve (EC) scalar multiplication for Elliptic Curve Diffie–Hellmann (ECDH) computation, Keyed-Hashing for Message Authentication (HMAC), AES-CBC-MAC-128 for the Message Integrity Code (MIC) computation, and AES-CTR-128 for symmetric encryption. These methods increase energy consumption and tend to be unsuitable for resource-constrained devices. Additionally, this solution does not consider the slow DoS attack [23], which is specific to the MQTT protocol. Although the performance parameters of MQTT have improved, there is no evidence of a security improvement, and the protocol still needs to be evaluated in IoT environments. Table 1 discusses the different existing security schemes with their advantages and disadvantages.

**Table 1.** Comparison of various existing works with their advantages and disadvantages.

| Works | Strategies | Description | Advantages | Disadvantages |
|---|---|---|---|---|
| Sadio et al. [24] | CHACHA20-Po1y1305 AEAD. | Proposed a lightweight security scheme based CHACHA20-Po1y1305 AEAD algorithm for MQTT/MQTT-SN. | High performance and provides dual authentication and encryption. | Less-secure and complex to implement. |
| Bogdanov et al. [25] | PRESENT. | Designed a lightweight structured cryptography protocol. | Minimum level of security and high performance. | Lacks providing security against various latest attacks. |
| Imdad et al. [26] | KSA-PRESENT. | Introduced key-scheduling-based PRESENT algorithm. | Enhances security and reduces the avalanche effect. | Algorithm complexity and overhead. |
| Diro et al. [27] | Lightweight security scheme in fog networks using EC. | Proposed a lightweight scheme with authentication and key management. | High security. | High computational cost of encryption algorithms. |
| Mektoubi et al. [28] | MQTT-RSA-ECC. | Presented ECC-based security without key revocation. | Medium security. | Poor performance, high delay, and increased energy consumption. |
| Singh et al. [16] | S-MQTT. | Developed secure MQTT with the assistance of key- and ciphertext-based attributes. | Acceptable level of performance with high security. | The high number of attributes increases the overhead and computational cost. |
| Wang et al. [29] | DP-ABE. | Permits two access control strategies and ensures high security. | High security and performance efficiency. | High overhead and computational cost. |
| Liao et al. [30] | SMQTT-ABE. | Developed a secure MQTT using improved ABE and chaos synchronization. | Medium level of security with improved attributes. | Lacks accomplishing a better trade-off between security and overhead. |

Sadio et al. [24] proposed using the Authenticated Encryption with Associated Data (AEAD)-CHACHA 20-POLY1305 algorithm as a scheme to secure restricted node communication through MQTT/MQTT-SN. The security scheme assumes the presence of a gateway between the publisher and the broker. The topic data are encrypted and authenticated successfully using a pre-shared secret key and CHACHA20-Po1y1305 AEAD. Although this

security framework uses a lightweight and efficient algorithm for message exchange and data encryption between the client and server, this security framework is less secure than other frameworks, such as TLS, and is not widely available. Furthermore, the CHACHA20-Poly1305 AEAD algorithm is relatively new, and adequate proof of security verification must be provided. The suitability of other symmetric encryption algorithms for MQTT communication has not yet been sufficiently evaluated in their work.

Bogdanov et al. [25] introduced a PRESENT algorithm, a lightweight structured cryptography protocol highly adapted to resource-limited network environments. It is a lightweight symmetric block cypher offering better IoT performance and security trade-offs. However, the linear functions of the PRESENT key-scheduling algorithm are used to identify the relationship among round keys, which leads to a slow and predictable bit transition. Therefore, Imdad et al. [26] introduced a PRESENT algorithm based on the Key-Scheduling Algorithm (KSA-PRESENT) scheme to solve these issues. The KSA-PRESENT scheme enhances the randomness and unpredictability of the round key for cryptographic security by combining the PRESENT-128 packet cypher with a more-complex non-linear function to generate the round key.

Diro et al. [27] proposed a fog-computing-based security scheme that relies on Elliptic Curve Cryptography (ECC) algorithms to provide secure communication for MQTT without compromising security. The proposed scheme exchanges secret keys securely using a reduced number of handshake messages. The ECC algorithm uses short private keys, which reduces the message sizes compared to other asymmetric cryptographic algorithms. However, it is unclear whether this scheme suits resources-constrained devices with limited resources. In their paper, Diro et al. analysed the performance of MQTT with the AES scheme following a DoS attack, considering communication and storage overhead and delays. AES provides a fast response, but it was found to have high memory consumption.

Many conventional solutions [31,32] rely on ABE algorithms instead of complex security schemes. For example, the RSA-based CP-ABE scheme was implemented in [33] with secret keys of a constant size and without using bilinear maps to reduce the complexity of RSA. However, this solution needs to address the key-revocation issue. Mektoubi et al. [28] proposed an ECC-based security scheme, but did not consider the key-revocation, scalability, and generality issues. The scheme increases delay and energy consumption, and the keystore used in this scheme degrades the performance of MQTT.

Singh et al. [16] developed Secure MQTT (SMQTT) by implementing Key Policy-Attribute-Based Encryption (KP-ABE) and CP-ABE separately using lightweight ECC. However, it does not focus on key revocation and group publish/subscribe. The ECC scheme uses a secret key for a long lifetime, which could weaken its security in a hostile environment. Additionally, as the number of attributes increases, the private key's length and the communication delay increase significantly. Mendoza-Cardenas et al. [17] intended to evaluate the performance of MQTT with CP-ABE to its adoption in an IoT environment. Wang et al. [29] developed a Dual-Policy-Attribute-Based Encryption model (DP-ABE) that allows two access control strategies and ensures high security in cloud scenarios. However, it increases the computational cost and overhead in the network. Deng et al. [34] presented a practical attribute-based access control scheme with CP-ABE support. It allows the Trusted Authority (TA) to handle user credentials efficiently. However, it meets single-point failure issues in many scenarios.

Liao et al. [30] developed a secure MQTT using improved ABE and chaos synchronisation. However, additional schemes in ABE-based MQTT may need to be better suited for devices with limited resources. The existing security schemes focus on either communication security or computational overhead. Finally, the CP-ABE-based security schemes could be more efficient for large-scale IoT applications. However, the number of attributes increases the key size and resource consumption, which are not highly fit to the resource-constrained IoT devices. Therefore, a lightweight authentication scheme for MQTT should be designed without sacrificing privacy and scalability while minimising computational overhead and cost.

## 3. The Proposed Scheme

### 3.1. Overview

The MQTT protocol is a lightweight application-layer protocol that requires lightweight security functions to improve the security of IoT communications. The proposed approach aimed to implement a CP-ABE scheme based on the PRESENT algorithm for the MQTT protocol. The main components of the proposed scheme are authentication using CP-ABE, PRESENT transmitting blind keys, self-key agreement using the previously accessed topic, and secure data publishing/subscribing in the IoT communication model. The MQTT broker must authenticate both MQTT clients, the publisher and subscriber, during data publishing/subscribing to secure the data communication.

The publisher and subscriber devices provide unique identities, such as a Universal Resource Identifier (URI), and their attributes to obtain prime numbers. In particular, the devices are responsible for executing the PRESENT algorithm to obtain the secret key by the CP-ABE scheme. By using this private key, publishers encrypt data and topics. After encrypting the data, the publisher hashes the attribute length using this key, known as the Attribute Length Key (ALK). The subscriber also executes this process while subscribing to a particular topic from the MQTT server, then named the self-key agreement at both the publisher and the subscriber. Figure 1 illustrates the ICP-ABE flow diagram.

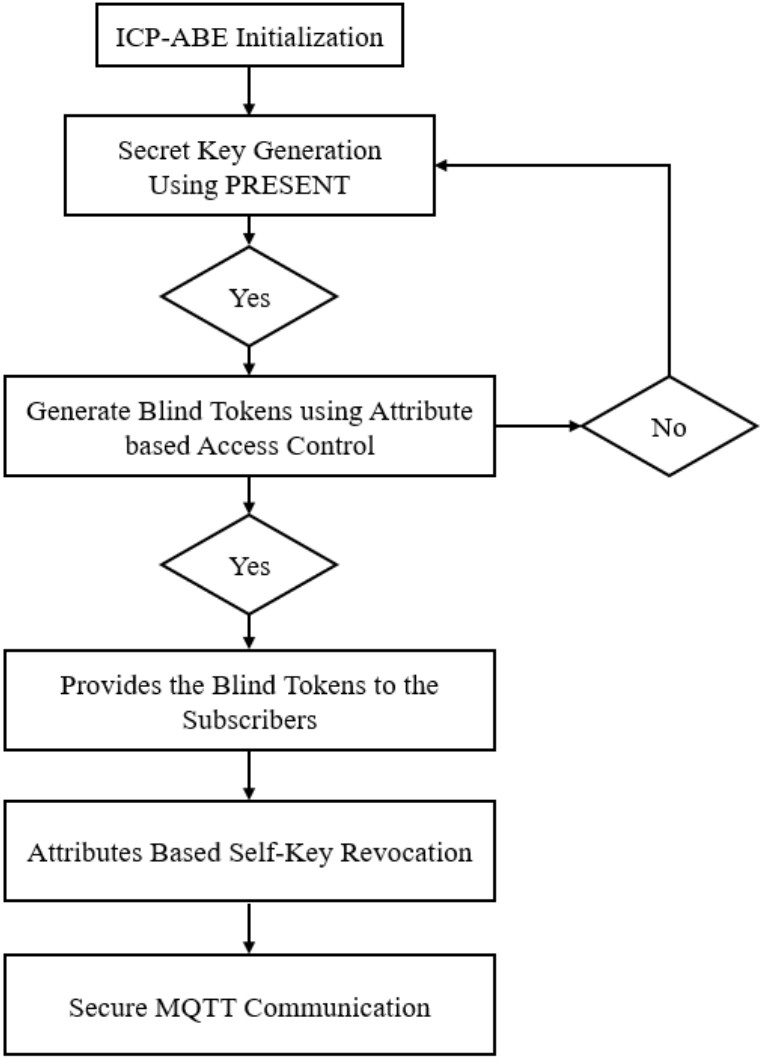

**Figure 1.** Flow diagram of ICP-ABE.

The published encrypted data are stored on the MQTT server with the index of the encrypted topic. When accessing a particular topic, the subscribing devices send a request message along with the encrypted topic using a secret key. The MQTT server sends a message to the corresponding subscriber by matching the encrypted topic in the index. The subscriber generates the ALK to decrypt the message received from the MQTT server. Finally, the proposed scheme updates the ALK using the previously accessed topic and key. This avoids secure key sharing among devices and ensures the security of the system. Due to the limited battery resources in IoT devices, the proposed scheme enables MQTT clients to store only the previous topic, secret key, and ALK.

### 3.2. Secret Key Sharing Using the PRESENT Algorithm

In most existing ABE schemes, the MQTT server generates secret keys based on attribute information about each user, which may compromise client privacy. The proposed scheme addresses this issue by separating the attribute-audit and key-extraction functions. CP-ABE allows the publisher to act as an attribute-audit centre. By limiting the secret key length using the key register rotation and attributes-based self-key revocation, the proposed ICP-ABE model minimises the overhead and resource consumption at the publishers. The overhead of publishers is nearly 64 bit, which is a minimum for the conventional works.

Publishers and subscribers have knowledge of the blind token and its associated attributes, while brokers are only authorised to know blind tokens without their corresponding attributes. Clients can submit their attributes and identities to the publisher through the broker during the data request. Publishers are responsible for auditing blind tokens according to the attributes and providing subscribers with encrypted data with a signature. Data encryption is performed using a blind key, and hashing is performed using ALK. Moreover, the blind token provides evidence that the client has specific attributes while keeping the attributes or contents of the token confidential. In particular, the blind token is initially shared using the secret key generated using the PRESENT algorithm. Figure 2 illustrates this process.

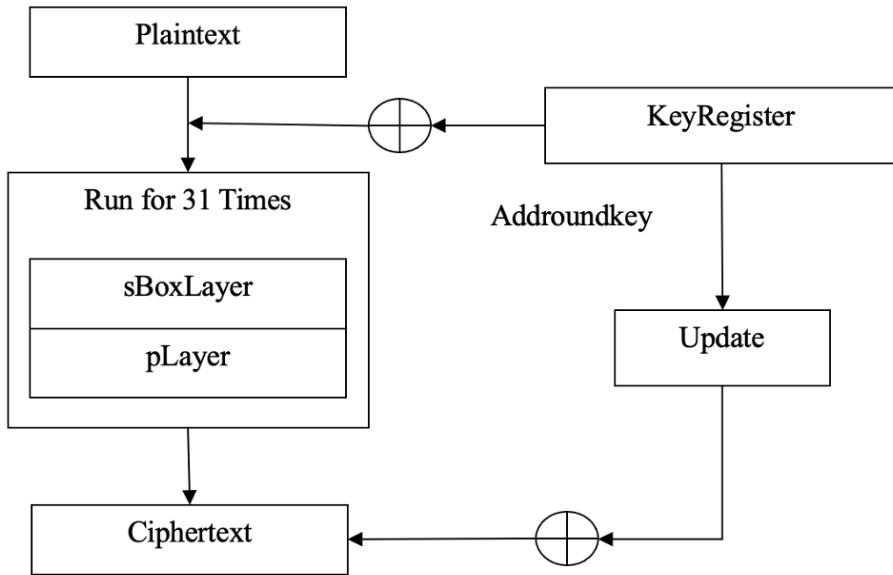

**Figure 2.** Functions in PRESENT algorithm.

The symbol ⊕ in Figure 2 denotes the XOR operation. The term key register field refers to the storage of keys for specific tokens, which rotates in every round. The private keys of the subscribers are stored in the key register for a specific time interval. The "all-rounder" encompasses key sets that are used to append the keys in every state. The S-box layer replaces the bytes in the array or sub-type forms, each consisting of four bits.

Initially, the subscriber sends a blind token subscriber content request to the publisher using MQTT for network initialization. The subscriber requests the publisher through the broker to send the content of interest, and the publisher is responsible for verifying the legitimacy of the blind token. The requested content is sent if the token is valid; otherwise, the request is aborted. The subscriber and the publisher store their attributes and relevant blind keys in the attribute set. The publishers audit the blind keys and associated attributes and, depending on the attribute policy, send the same blind key to the broker to deliver the requested content to the subscriber. The blind key is securely sent using PRESENT encryption [28], which generates a secret key by running 31 rounds of the XOR operation with a key size of 80 bit. PRESENT consists of the following functions:

1.  AddRoundKey, which adds a round key with each state.
2.  The S-box layer replaces each byte in an array with a sub-byte (the S-box is of type 4 bit to 4 bit).
3.  The pLayer, which permutes each state into the predefined position.

The user-provided secret keys are stored in the key schedule, and a key register is rotated 61 positions to the left. In the S-box, the leftmost four bits constitute the key. When the requested content is sent, both the publisher and the subscriber independently modify the blind key by executing the XOR operation between the previous blind key, the blind token, and the previously accessed topic.

### 3.3. Attribute-Based Signature Scheme and Self-Key-Revocation Scheme Using Attributes

After encrypting the data, the publisher calculates the attribute length using ALK, which is also used to hash the secured data. This process is also executed in the subscriber when subscribing to a particular topic on the MQTT server. The second component of the scheme is called the self-key agreement, which both the publisher and the subscriber perform. The publisher stores the published encrypted data on the server with the index of the encrypted topic. Subscribing devices send a request message and an encrypted topic using a blind key to access a particular topic. When the server finds a match in the index, it sends a message to the subscriber associated with the encrypted topic. To decrypt the received message, the subscriber generates the ALK. The proposed scheme updates the ALK using the previously accessed topic and key. This approach avoids secure key sharing between devices and ensures the security of the system. To accommodate devices with limited battery resources, the proposed scheme allows MQTT clients to store only the previous topic, blind key, and ALK. This allows them to generate the security key on their own. The KeepAlive parameter alone is insufficient to prevent slow DoS specific to the MQTT protocol. Therefore, the proposed scheme disables client nodes from defining the KeepAlive parameter on the server. Instead, the broker represents a common KeepAlive parameter value for all client nodes. This approach ensures lightweight authentication and privacy protection for IoT clients.

## 4. Security and Computational Cost Analysis

In this section, we performed a security and computational cost analysis of the proposed scheme, which aimed to address some of the challenges associated with applying CP-ABE schemes over MQTT using lightweight schemes. The symbols used in ICP-ABE are defined in Table 2.

### 4.1. Security Analysis

We considered the following types of attack on MQTT communication: DoS, Spam, Poison, and Man-in-the-Middle (MitM) attacks. DoS attacks attempt to waste network resources by unnecessarily flooding control messages. Botnet groups can also infect clients and launch various attacks, such as DoS, Poison, and Spam, on MQTT communication. MitM attackers can modify MQTT packets between servers and clients, reducing their efficiency. Additionally, the chosen-ciphertext attack attempts to identify plaintext by selecting certain ciphertexts.

**Table 2.** Symbols and definitions.

| Symbol | Definition |
|---|---|
| $O()$ | Complexity. |
| $N$ | Number of attributes. |
| $H$ | The complexity of the message and ciphertexts. |
| $1O$ | The complexity of the XOR operation. |
| $Time_{Comp}$ | Time complexity. |
| $R_q R_s$ | The sum of the round-trip time taken by topic requests $R_q$ and the response from the server $R_s$. |
| $K_s$ | Key Size. |
| $PF(t)$ | Failure probability of key tracing. |
| $PS(t)$ | Success probability of key tracing. |
| $t$ | Time. |
| $K_x(t)$ | Number of keys that an attacker tries to process before t. |
| $2^n$ | Total number of possible keys. |
| $n$ | Number of input bits. |
| $m$ | Number of output bits. |
| $a0$ | Number of zeros in the n-bit sequence of the secret key generation of the PRESENT algorithm. |
| $a1$ | Number of ones in the n-bit sequence of the secret key generation of the PRESENT algorithm. |
| $tn$ | Number of attributes |
| $k$ | The average size of an attribute. |
| $l$ | Number of rows in the access structure. |

### 4.1.1. Security against Slow DoS Attack

In general, a robust cryptographic mechanism in conjunction with server support can prevent attackers from accessing the blind token and modifying messages between smart sensors and the server. However, a slow DoS attacker can copy a valid message and slowly re-send it to the server, extending the KeepAlive parameter value and potentially gaining network access. To seize all available connections to the broker, the slow DoS attacker establishes a high number of connections with minimal resource consumption. If the attacker establishes all available MQTT connections, this leads to a DoS attack, which can negatively impact MQTT performance, even when the KeepAlive parameter is used. Implementing the ICP-ABE scheme in MQTT prevents this attack since the server determines the common KeepAlive parameter and all clients revoke the key independently, preventing the attacker from sending packets again for a long period. ALK values are generated in a predetermined manner and vary during communication. If a node resends a packet using the hash value of the encrypted message with the old ALK, the server and client can identify attack packets based on mismatched hash values.

### 4.1.2. Security against Spam and Poison Attack

The proposed ICP-ABE scheme provides protection against Spam and Poison attacks in MQTT communication. The Spam attacker sends unsolicited commercial messages to many clients, which can have a significant impact on a group of clients who have published similar content. However, due to the differences in published topics between clients in MQTT, it is not always possible to implement the Spam attack broadly. On the other hand, Poison attacks aim to inject erroneous data into the IoT network. Such an attack is possible when the attacker compromises the publishers and obtains secret information. To prevent

these attacks, the proposed ICP-ABE scheme uses different secret keys, such as a secret key, blind key, and ALK, in various steps to ensure MQTT communication security.

### 4.1.3. Security against MitM Attack

MitM attacks are capable of breaking security schemes, which makes them a significant threat to MQTT communication. Once an attacker has compromised nodes, he/she can execute various MitM attacks. Although key revocation is often effective in preventing MitM attacks, it may not always be possible to use it. Furthermore, the PRESENT algorithm used in the proposed scheme has the disadvantage that it is susceptible to biased input in the keyspace, which can be exploited by attackers. If an attacker obtains the secret key, he/she can apply a brute-force method to trace the blind and ALKs, compromising the strength of the proposed scheme.

### *4.2. Computational Complexity Analysis*

To estimate the computational cost of the proposed ICP-ABE scheme, we considered the operations involved in secret key generation, blind key generation, and Pub/Sub messages using the PRESENT algorithm.

### 4.2.1. Secret Key Generation and Blind Key Sharing

The PRESENT algorithm executes the XOR operation for 31 rounds and generates the round key for blind key sharing. Additionally, it performs shifting and permutation in the S-box layer and pLayer, respectively. Assuming that the cost of performing an XOR operation between two bytes and one-byte rotation is equal, the cost of applying XOR between two bytes is denoted $X$, and one-byte rotation is denoted $R = 1O$. Similarly, a table lookup in the S-box is denoted as $1L$ for one byte, and the cost of two-byte multiplication is denoted as $1M$. For shifting operations, it experiences $31O$ for 31 rounds, and for permutation, it experiences $6O$. Additionally, it performs 12 XOR operations ($12O \times 4$) for a 4 x 4 matrix, resulting in $48O$. The key schedule executes 16 transformations and generates an additional $64M$ confusion. Furthermore, the communication complexity is determined by the length of the ciphertext, that is the blind encrypted key by a secret key. During blind key sharing, the length of the key size is $H$ bits. Therefore, the computational complexity of the algorithm is as follows.

$$O(\text{PRESENT}) = 85O + 64M + H \tag{1}$$

### 4.2.2. Secure Pub–Sub Messages

In most existing schemes, the size of the secret key is determined on the basis of the number of attributes used in the scheme. However, in the proposed scheme, the secret key is generated through an XOR operation between the blind key, the blind token, and the topic previously accessed. In the case of the generation of ciphertext involving attributes $N$, the complexity can be determined as follows.

$$O(\text{Ciphertext}) = H + H + N * 2H \tag{2}$$

The complexity depends on the size of both the message and the ciphertext and assumes that they are equal, denoted by $H$. However, the proposed scheme reduces the computational complexity by using a self-key revocation scheme. The complexity of the XOR operation is added as $1O$.

$$O(\text{Ciphertext}) = H + H + 1O \tag{3}$$

Thus, the computational complexity of the proposed scheme is represented as follows.

$$O(\text{ICP-ABE}) = 86O + 64M + 3H \tag{4}$$

*4.3. Time Complexity*

The MQTT request–response and data communication metrics were used to estimate the time complexity of ICP-ABE. The duration of the request–response is defined as the time it takes for a client to send a request for a topic and receive a response. The publisher acknowledges the client's request through a set of attributes, while the server acknowledges the request through a blind key. The server sends a response after verifying the client node. The time complexity, $Time_{Comp}$, is calculated at the end of the client using the following equation.

$$Time_{Comp} = R_q R_s + T_{Communication} \qquad (5)$$

$R_q R_s$ can be estimated as the sum of the round-trip time taken by topic requests $R_q$ and the response from the server $R_s$. Furthermore, the time taken to deliver the data is added to $R_q R_s$ to calculate the overall time complexity of the proposed work. However, in the existing scheme [16], $R_q R_s$ depends on the number of attributes and the length of the secret key, leading to a significant increase in time complexity. In contrast, the proposed scheme generates the blind key differently, rather than based on attributes, resulting in a considerable reduction in time complexity compared to existing works.

## 5. Evaluation and Validation

*5.1. Provable Security*

In the proposed ICP-ABE scheme, MQTT communication is secured by the PRESENT algorithm, and blind tokens are used to generate blind keys for secure data transfer. The security strength of the scheme was evaluated based on the key size of the PRESENT algorithm and the attribute-based encryption scheme. The success probability and the failure probability of key tracing in a cryptographic algorithm with a key size of $K_s$ are represented by $PS(t)$ and $PF(t)$, respectively. The number of keys that an attacker tries to process before time $t$ is denoted by $K_x(t)$, and the total number of possible keys is $2^n$.

$$PF(t) = 1 - PS(t) \qquad (6)$$

$$PS(t) = K_x(t)/2^n \qquad (7)$$

By combining (6) and (7), we obtain:

$$PF(t) = 1 - (K_x(t)/2^n) \qquad (8)$$

The PRESENT algorithm provides better security against brute-force and MitM attacks compared to the ECC algorithm used in [16]. However, according to [28], the RSA-based solution offers a better security strength. The security strength of ICP-ABE involves several critical factors, such as the size of the key, the efficiency of a random function, the relationship factor between bits, and the attribute–ciphertext relationship factor. PRESENT generates subkeys randomly, and the proportion of zeros and ones determines the efficiency of the subkeys produced by the PRESENT algorithm used in ICP-ABE. The purpose of measuring the efficiency of random functions is to determine whether the number of ones and zeros is equal to or greater than 50% during the generation of subkeys. If the generated subkeys do not meet these conditions, PRESENT does not meet the basic characteristics of randomness. Therefore, the generation of subkeys in the PRESENT algorithm weakens the security strength of ICP-ABE. In Equation (9), we define the Efficiency of a Random Function (ERF), where $a0$ and $a1$ represent zeros and ones, respectively, in a sequence of bits $n$ during the generation of subkeys in the present algorithm.

$$ERF = \frac{(a0 - a1)^2}{n} \qquad (9)$$

The existing work in [16,28] investigated a hybrid of the Rivest–Shamir–Adleman (RSA) and the Elliptic Curve Digital Signature Algorithm (ECDSA) and ECC algorithms,

respectively. However, the usage of large secret keys by RSA leads to an inadequate ERF and compromised security due to the multiplication of two prime numbers.

The second criterion for security strength estimation is the bit-relationship test of ICP-ABE, which verifies its bit confusion and diffusion properties. In this test, the $n$ input bits are mapped onto the $m$ output bits. The bit-relationship function is denoted as follows.

$$RF(\text{Plaintext})^n = RF(\text{Ciphertext})^m \qquad (10)$$

Note that $m$ is not necessarily equal to the $n$ bits. However, if $m \ll n$, it reduces the efficiency of $RF$, while increasing the security strength of ICP-ABE against cryptanalysis attacks such as MitM. In [28], RSA $RF(\text{Plaintext})^n$ was used, where the length of the prime numbers $p * q$ was not equal to the security strength of the bit length $n$. A poor bit-relationship function can allow attackers to breach the security scheme.

The attribute–ciphertext relationship factors indicate the impact of the number of attributes on the length of the ciphertext. The length of the ciphertext affects the network performance in terms of delay and throughput, while a short length facilitates the key tracing by attackers. Here, $tn$ represents the number of attributes, $k$ denotes the average size of an attribute, and $l$ represents the number of rows in the access structure.

$$\text{Ciphertext Length} = \frac{n}{m} + (tn * k) + l \qquad (11)$$

In ICP-ABE, the impact of $tn * k$ on the ciphertext length is negligible because ALK-based hashing always has the same length for any input data length, i.e., encrypted message. The proposed scheme only considers the length of the attribute in the generation of a key, but does not affect the creation of a ciphertext. Therefore, the second term, $tn * k$ in (11), is null for the proposed scheme. The negative impact of RSA [16] and ECC [28] on the security strength of the attribute-based encryption scheme is higher compared to that of ICP-ABE. The following Table 3 compares the security strength of the proposed ICP-ABE with four existing methods, which are simple PRESENT, KSA-PRESENT, RSA-ECC, and S-MQTT.

**Table 3.** Attack prevention ability of various algorithms against different attacks.

| Security Breaches | Attack Prevention Ability of Algorithms | | | | |
|---|---|---|---|---|---|
| | **ICP-ABE** | **Simple PRESENT** | **KSA-PRESENT** | **RSA-ECC** | **S-MQTT** |
| DoS | High | High | High | High | High |
| Slow DoS | High | No | No | Medium | No |
| Spam | High | Medium | High | Low | High |
| Poison | High | Medium | High | High | Low |
| MitM | High | Medium | High | Medium | Medium |

*5.2. Simulation Results*

As discussed earlier, the MQTT protocol was extended to include PRESENT, the ALK, and self-key revocation, and validation was performed using the JcrypTool, Scyther, and Cooja simulators. The proposed ICP-ABE model ran on Ubuntu 18.04 LTS with an Intel i3 2.5 GHZ CPU and 4 GB memory. Our results, shown below, indicated that ICP-ABE performed well with and without attacks. The proposed work can be considered a strong and lightweight security scheme for MQTT in an IoT environment. To evaluate the performance of ICP-ABE and the existing simple PRESENT [25], KSA-PRESENT [26], RSA-ECC [28], and SMQTT [16] schemes, simulations on IoT nodes were performed using Ubuntu 14.04 LTS 64 bit and Contiki-3.0. Table 4 presents the simulation parameters.

**Table 4.** Simulation model.

| Parameters | Values |
| --- | --- |
| Application Protocol | MQTT |
| Total Number of Nodes | 31 (Router-1, Publisher-10, and Subscriber-20) |
| Number of Attacker Nodes | 4 |
| Dos Attack Nodes | 2 |
| MitM Attack Nodes | 2 |
| Simulation Area | 100 m × 100 m |
| Transmission Range | 50 m |
| Simulation Time | 5 min |
| MQTT-Broker | Mosquitto-rsmb broker-1.3.0.2 |
| Algorithms | ICP-ABE, simple PRESENT, KSA-PRESENT, RSA-ECC, and S-MQTT |

Average energy consumption: This measures the average amount of energy consumed by the MQTT client on the network when performing authentication and publishing or subscribing to messages from the MQTT broker.

CPU energy consumption: This measures the average amount of energy consumed by the CPU in the MQTT client during the execution of the authentication and communication processes.

Execution time: This is the total time taken to execute the entire process of a cryptography algorithm.

Computation overhead: This is the time taken to perform the computations of the algorithm.

Communication overhead: This is defined as the length of the ciphertext, which includes all additional information added to the original plaintext to secure it.

Strength evaluation criteria: This is defined as the ratio between the length of the ciphertext and the plaintext, which is used to evaluate the strength of the security scheme.

*5.3. Results Analysis*

Table 5 shows the performance results of several scenarios running under different scenarios.

CPU and Avg energy consumption: Figures 3 and 4 illustrate the results of the CPU and the average energy consumption for ICP-ABE, simple PRESENT, KSA-PRESENT, RSA-ECC, and SMQTT, respectively. The proposed scheme employs blind key generation, which reduces the length of both the secret key and the ciphertext. As a result of the reduced key size, ICP-ABE reduces the CPU load and consumes minimal energy to identify and authenticate IoT devices. Figure 3 shows that the CPU energy consumption of ICP-ABE was much lower than that of other existing methods. For example, the proposed ICP-ABE consumes 0.0000409 and 0.00464 joules for the MitM and DoS attack scenarios in Figure 3. Similarly, the average energy consumption of ICP-ABE is 0.00155 and 0.00164 joules for the MitM and DoS attack scenario in Figure 4. Unlike ICP-ABE, the simple PRESENT and KSA-PRESENT algorithms do not consider the secret key length for key revocation. Thus, this significantly increases the resource consumption at the nodes. Compared to the proposed scheme, the RSA-ECC scheme had the additional disadvantage of the keystore used in the existing work, which degrades the protocol performance. Revocation of the self-key reduces the overall energy consumption by sharing the secret key separately.

MitM attacks can compromise the key values of legitimate nodes. By setting the value of the KeepAlive parameter to high, attackers may be able to send unnecessary control messages and request messages to the server. When used in the context of the IoT, SMQTT tends to result in unnecessary energy consumption due to the length of the secret key, which

increases with the number of attributes. Based on the simulation results, the proposed scheme reduces both CPU consumption and average energy consumption more effectively than RSA-ECC and SMQTT.

**Table 5.** Results of various existing works with and without attacks.

| | | CPU Energy Consumption | Average Energy Consumption | Execution Time | Computational Overhead | Strength Evaluation Criteria | Communication Overhead |
|---|---|---|---|---|---|---|---|
| ICP-ABE | Without Attacks | 0.0000256 | 0.001423 | 0.4 | 295.741 | 1.2 | 64 |
| | MitM | 0.0000409 | 0.00155 | 1.6 | 305.49 | 1.2 | 64 |
| | DoS | 0.0000464 | 0.00164 | 2 | 315.1 | 1.2 | 64 |
| Simple PRESENT | Without Attacks | 0.0000312 | 0.00148 | 1.5 | 285.93 | 0.399 | 80 |
| | MitM | 0.0000478 | 0.00166 | 2 | 324.05 | 0.399 | 80 |
| | DoS | 0.000048 | 0.00169 | 3.8 | 335.65 | 0.399 | 80 |
| KSA-PRESENT | Without Attacks | 0.0000322 | 0.00153 | 0.9 | 305.489 | 0.399 | 80 |
| | MitM | 0.000047 | 0.0016 | 1.8 | 329.57 | 0.399 | 80 |
| | DoS | 0.0000473 | 0.00166 | 2.4 | 330.657 | 0.399 | 80 |
| RSA-ECC | Without Attacks | 0.0000331 | 0.00158 | 2.5 | 310.96 | 1 | 128 |
| | MitM | 0.000266 | 0.00175 | 3 | 330.25 | 1 | 128 |
| | DoS | 0.000298 | 0.00177 | 3.65 | 350.21 | 1 | 128 |
| SMQTT | Without Attacks | 0.000038 | 0.00161 | 2.1 | 315.24 | 1 | 128 |
| | MitM | 0.000472 | 0.00177 | 2.8 | 355.55 | 1 | 128 |
| | DoS | 0.000486 | 0.00185 | 3.8 | 362.36 | 1 | 128 |

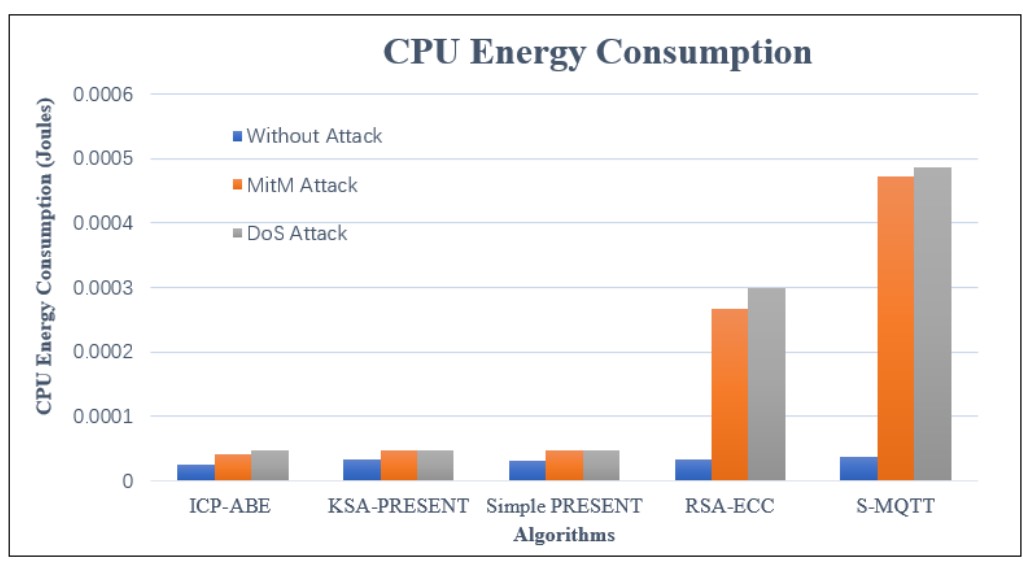

**Figure 3.** CPU energy consumption.

Execution time and computational overhead: Figures 5 and 6 illustrate the execution time and computational overhead results of the ICP-ABE, simple PRESENT, KSA-PRESENT, RSA-ECC, and SMQTT algorithms. The results were obtained for the scenarios without attack, the MitM attack, and the DoS attack. From the figures, the execution time and computational overhead of the proposed ICP-ABE was less compared to the other protocols. The reason behind this is that the proposed model considers the advantages of the PRESENT and ABE models in blind token generation and self-key revocation. Thus, it simplifies the algorithm process of ICE-ABE and optimises the lightweight design without compromising

the security level and performance efficiency of MQTT. For example, ICP-ABE, simple PRESENT, KSA-PRESENT, RSA-ECC, and SMQTT achieve 1.6, 1.8, 2.3, and 2.8 s execution times, respectively, for the MitM attack scenario in Figure 5. Similarly, the computational overhead of ICP-ABE, simple PRESENT, KSA-PRESENT RSA-ECC, and SMQTT is 315.10, 330.66, 335.65, 350.21, and 362.36 ms, respectively, to detect DoS attacks in Figure 6. This is due to the lightweight design of ICP-ABE compared to the other ones.

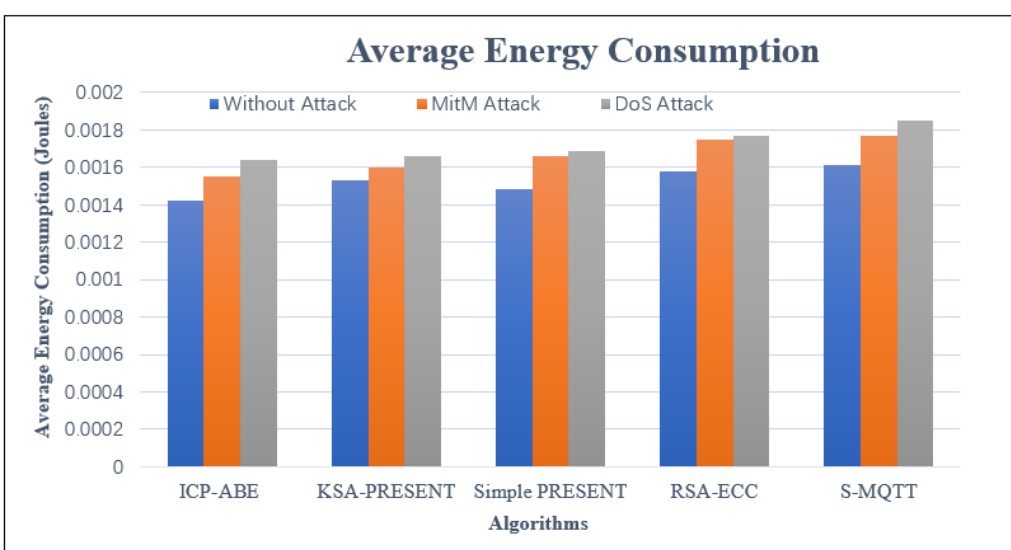

**Figure 4.** Average energy consumption.

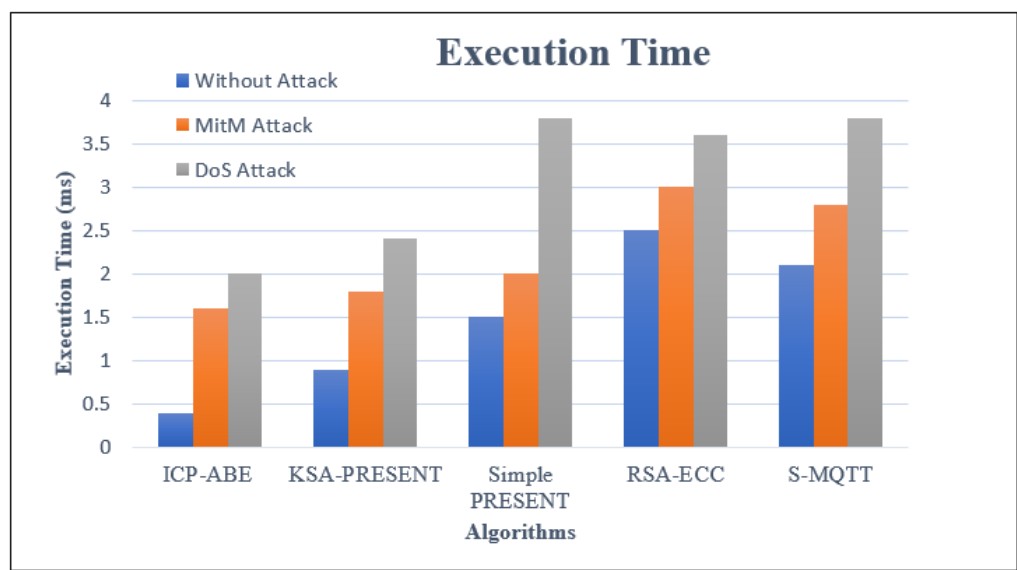

**Figure 5.** Execution time.

Strength evaluation criteria and communication overhead: Figures 7 and 8 illustrate the evaluation of the security strength and communication overhead for ICP-ABE, simple PRESENT, KSA-PRESENT, RSA-ECC, and SMQTT in normal and vulnerable environments, including DoS and MitM attacks. The length of the ciphertext is a crucial factor in determining the suitability of a security scheme for IoT devices. However, unlike the proposed scheme, both existing schemes increase the length of the ciphertext as the number of attributes increases, resulting in degraded security strength. For example, under the MitM and DoS attack scenarios, the security strength of ICP-ABE was 1.2. Although the existing work satisfied one of the security strength criteria for all scenarios due to its key

scheduling and attribute-based ciphertext-creation method, its communication overhead also increased as a result of the longer length of the ciphertext.

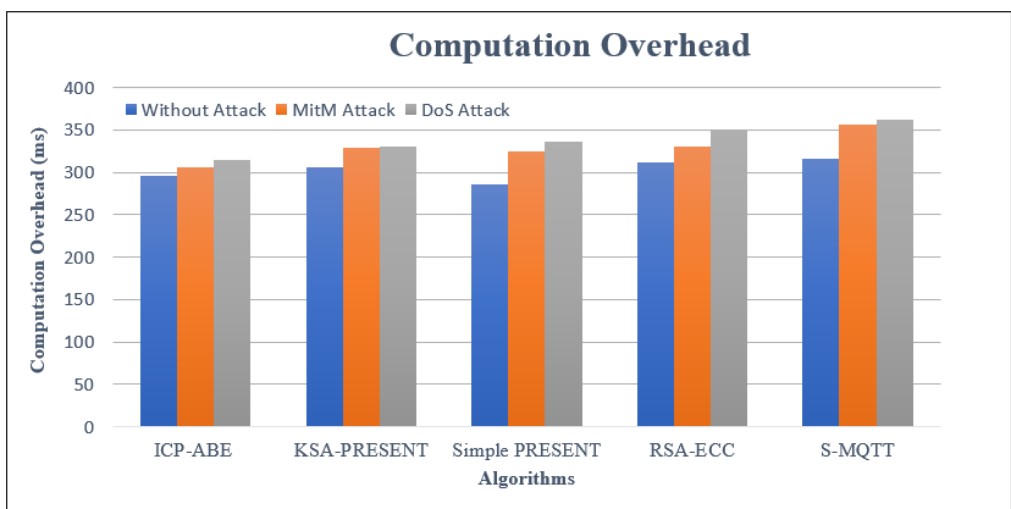

**Figure 6.** Computational overhead.

Furthermore, the communication overhead of simple PRESENT, KSA-PRESENT, RSA-ECC, and SMQTT was high in all scenarios compared to the proposed scheme. This is because the length of the secret key increases with the number of attributes, resulting in unnecessary communication overhead and poor strength when used in an IoT environment. For example, the communication overhead of ICP-ABE is only 64 bit in both the normal and attack scenarios in Figure 8. The simple PRESENT and KSA-PRESENT accomplished 80 bit of communication, and RSA-ECC and S-MQTT achieved 128 bit in communication overhead in all scenarios due to their key scheduling and attribute-based ciphertext-creation process.

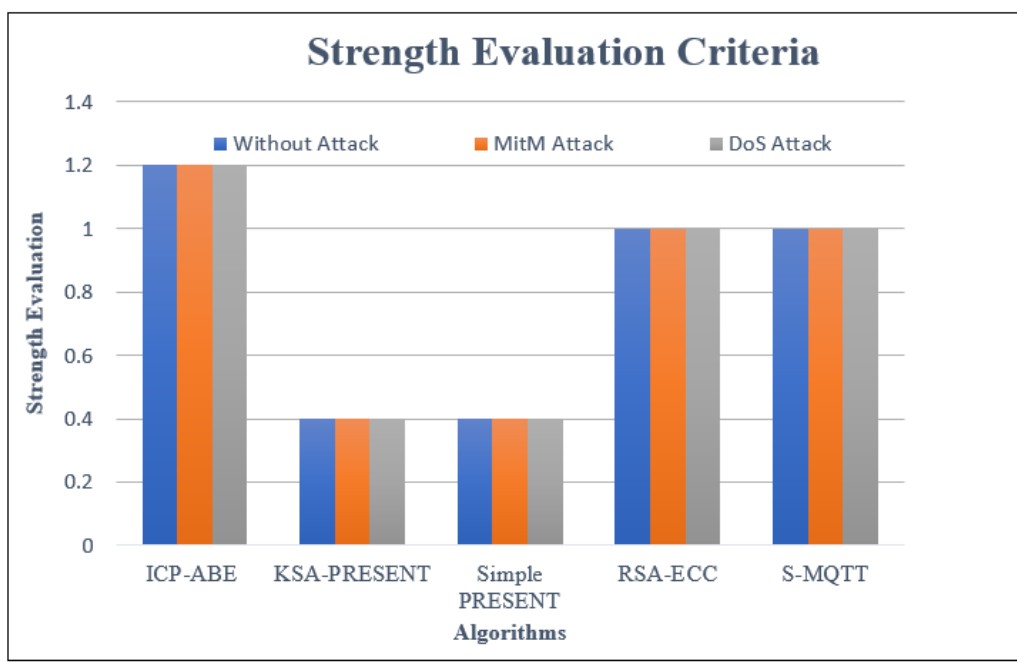

**Figure 7.** Strength evaluation criteria.

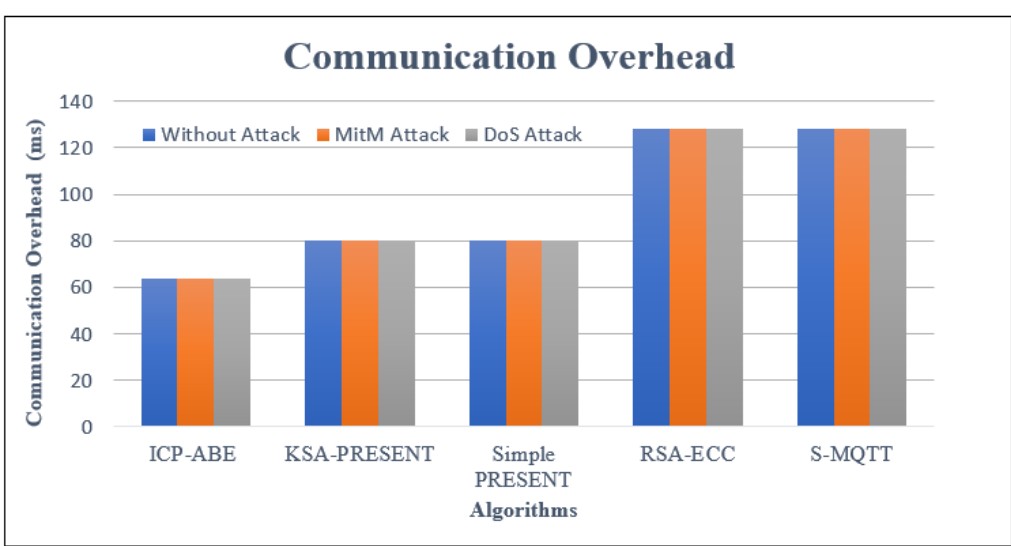

**Figure 8.** Communication overhead.

## 6. Conclusions

This work presented a lightweight ICP-ABE scheme for secure MQTT communication, which was based on a previously proposed scheme. The proposed scheme leverages attribute separation and blind keys associated with attributes to reduce the computational complexity of the CP-ABE scheme. Furthermore, we evaluated and compared the proposed scheme with four existing schemes, simple PRESENT, KSA-PRESENT, RSA-ECC, and SMQTT, using provable security and formal analysis methods. The simulation results indicated that ICP-ABE significantly improved the performance and security of MQTT while exhibiting lightweight and secure characteristics in both standard and attack scenarios.

However, despite our efforts, there are still some issues with the ICP-ABE scheme that we aim to address in future work, such as the revocation of malicious users and tracking of malicious users with similar attribute sets. We will continue to enhance the security of the ICP-ABE solution and explore its applicability to other application layer protocols in the IoT. Additionally, we plan to evaluate the results of the improved ICP-ABE solution against existing lightweight security solutions and implement them on a natural healthcare IoT platform. Our future work will also focus on enhancing the feasibility of secure communications for IoT devices based on the Pub–Sub architecture.

**Author Contributions:** Conceptualization, methodology, software, validation, formal analysis, data curation, and writing—original draft preparation, S.T.; writing—review and editing, supervision, V.G.V. All authors have read and agreed to the published version of the manuscript.

**Funding:** This research received no external funding.

**Data Availability Statement:** No new data were created or analyzed in this study.

**Conflicts of Interest:** The authors declare no conflict of interest.

## Abbreviations

The following abbreviations are used in this manuscript:

| | |
|---|---|
| ABE | Attribute-Based Encryption |
| AEAD | Authenticated Encryption with Associated Data |
| ALK | Attribute Length Key |
| CP-ABE | Ciphertext Policy-Attribute-Based Encryption |
| EC | Elliptic Curve |
| ECC | Elliptic Curve Cryptography |
| ECDSA | Elliptic Curve Digital Signature Algorithm |
| ECDH | Elliptic Curve Diffie–Hellman |

| ERF | Efficiency of a Random Function |
| HMAC | Keyed-Hashing for Message Authentication |
| ICP-ABE | Improved Ciphertext Policy-Attribute-Based Encryption |
| IoT | Internet of Things |
| KP-ABE | Key Policy-Attribute-Based Encryption |
| KSA | Key Scheduling Algorithm |
| MQTT | Message Queuing Telemetry Transport |
| MIC | Message Integrity Code |
| MitM | Man-in-the-Middle |
| RSA | Rivest–Shamir–Adleman |
| SMQTT | Secure MQTT |
| TLS | Transport Layer Security |
| URI | Universal Resource Identifier |
| XOR | Exclusive OR |

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
