# Peer review of "On the Efficiency of a Lightweight Authentication and Privacy Preservation Scheme for MQTT"

_electronics, doi:10.3390/electronics12143085_

Round 1
Reviewer 1 Report
In this work, the authors propose an enhanced approach by integrating Improved Ciphertext-Policy Attribute-Based Encryption (ICP-ABE) with a lightweight symmetric encryption scheme, namely PRESENT, to enhance the security of MQTT. The utilization of the PRESENT algorithm allows for the secure sharing of blind keys among clients, thereby strengthening the overall security of the system. My concerns are as follows
1. without a notation table, it is hard to read the paper.
2. what is the meaning of various symbols in Figure 1? Explain Figure 1 in some detail like update, key register, etc.
3. what is novel in the paper, add the novelty and contribution section in the paper.
4. Explain the equation in some detail like 6, 7 , 8 , and 9.
N/A
Author Response
Reviewer 1:
1) Without a notation table, it is hard to read the paper.
Authors’ response:
Thank you for this comment. We have added a table of symbols and definitions in Section 4 to supplement the meaning of the formulae in the proof parts of Sections 4 and 5.
2) What is the meaning of various symbols in Figure 1? Explain Figure 1 in some detail like update, key register, etc.
Authors’ response:
In Section 3.2, we added more information about the specific explanation of the PRESENT algorithm used in this scheme. These contents have been highlighted in the revised manuscript.
3) What is novel in the paper, add the novelty and contribution section in the paper.
Authors’ response:
The update on the novelly and contributing parts of the ICP-ABE scheme presented in the paper has been presented in Section 1. The updated parts have been highlighted in the Contributions part of Section 1.
4) Explain the equation in some detail like 6, 7 , 8 , and 9.
Authors’ response:
We have included detailed equations 6 - 9 in the revised manuscript. The added part of the description is marked by highlighting in section 5.1. In addition, a table of the symbols and definitions used in the formulae has been added to the new manuscript in Section 4, which might be more helpful in understanding the formulae.
Reviewer 2 Report
CP-ABE-based security scheme is enhanced using a PRESENT algorithm. However, the over head of PRESENT is not taken into account when evaluating.
In the improved scheme, clients act as an attribute audit center. how much over head is that on a resource constrained IoT devices (which is the main objective of the work)? Analysis of this is required to see any merits of the proposed improvements.
The computational over head of the proposed method should be analyzed in more details and more results are required to justify the method.
Author Response
Reviewer 2:
1) CP-ABE-based security scheme is enhanced using a PRESENT algorithm. However, the over head of PRESENT is not taken into account when evaluating.
Authors’ response:
Thanks for your comments to help us improve the quality of our evaluation in this new version. We added more comparators to the original, such as Simple-PRESENT and KSA-PRESENT. We also added simulations of these schemes in terms of execution time and computational overhead. These results and tables are shown in Section 5.3.
2) In the improved scheme, clients act as an attribute audit center. how much over head is that on a resource constrained IoT devices (which is the main objective of the work)? Analysis of this is required to see any merits of the proposed improvements.
Authors’ response:
Section 3.2 describes the improved solution in detail and gives the advantages. In Section 5.3, we analyse and discuss the ICP-ABE scheme by evaluating the improved scheme against several existing schemes in simulations and show that the ICP-ABE scheme outperforms the other schemes in terms of security and overhead.
3) The computational over head of the proposed method should be analyzed in more details and more results are required to justify the method.
Authors’ response:
The computational overhead of the proposed work is analysed with four existing works and different attack scenarios, which are included in Section 5.3. The specific simulation results and analysis have been highlighted.
Reviewer 3 Report
The authors propose an encryption method, namely ICP-ABE, to improve the security of the MQTT protocol. The method makes use of a cryptography algorithm illustrated by Katagi et al. in 2008. The authors evaluate the efficiency of the scheme and compare the proposed method with two other solutions from the perspective of energy consumption and communication overhead. The main contribution, i.e. the proposed scheme, is uncertain, due to the fact that the scheme is based on two well-known algorithms. However, the topic is interesting. As follows, several issues are depicted.
- At first, the state-of-the-art analysis misses several works. Few works are analyzed, and the main disadvantages and advantages of them should be illustrated.
- A summary table of the state-of-the-art works could improve the description of the advantages of the proposed scheme, compared to the works of the community.
- The proposed scheme needs a flow diagram. The figure can help the comprehension of the new scheme and all the phases.
- The main contribution should be better explained. What is the contribution of the authors to the new proposed scheme? How have the two algorithms been improved?
- The compared factors in section 5.1. can be illustrated in a table.
- The simulation setup misses the specs of the hardware used for the simulations.
- A table can replace Figure 5. Moreover, more graphs can be included; for example, which types of attacks have the greatest impact?
Minor editing of English language required
Author Response
Reviewer 3:
1) At first, the state-of-the-art analysis misses several works. Few works are analyzed, and the main disadvantages and advantages of them should be illustrated.
Authors’ response:
The additional part on the analysis of the advantages and disadvantages of the existing work has been described in Section 2.
2) A summary table of the state-of-the-art works could improve the description of the advantages of the proposed scheme, compared to the works of the community.
Authors’ response:
In Table 1, we show a comparison of the advantages and disadvantages of the various existing efforts mentioned in Section 2.
3) The proposed scheme needs a flow diagram. The figure can help the comprehension of the new scheme and all the phases.
Authors’ response:
We have provided a diagram of how the ICP-ABE scheme is based on the MQTT protocol for secure communication. It is included in Figure 1.
4) The main contribution should be better explained. What is the contribution of the authors to the new proposed scheme? How have the two algorithms been improved?
Authors’ response:
Section 1 provides a more detailed description of how we have used the PRESENT algorithm and the CP-ABE scheme to achieve secure communication in resource-constrained devices in the IoT.
5) The compared factors in section 5.1. can be illustrated in a table.
Authors’ response:
We have added a table on the evaluation of the ability of the various algorithms used in the simulations to prevent different attacks. The comparisons are included in Section 5.1 with Table 3.
6) The simulation setup misses the specs of the hardware used for the simulations.
Authors’ response:
We have highlighted the hardware execution environment set up and applied in the Cooja simulator in Section 5.2.
7) A table can replace Figure 5. Moreover, more graphs can be included; for example, which types of attacks have the greatest impact?
Authors’ response:
Table 5 in Section 5.3 shows the performance results data for these evaluation scenarios with and without the attack state. In addition, since new evaluation scenarios were added to the original experiments, we replaced the specific evaluation figures (Figure 3 - Figure 8).
Round 2
Reviewer 2 Report
Thank you for addressing my concerns from the previous version.
Reviewer 3 Report
The authors answered all my previous comments exhaustively. The paper's quality has been improved and is now ready for publication.